# In Vitro Qualitative Evaluation of Root-End Preparation Performed by Piezoelectric Instruments

**DOI:** 10.3390/bioengineering9030103

**Published:** 2022-03-02

**Authors:** Calogero Bugea, Federico Berton, Antonio Rapani, Roberto Di Lenarda, Giuseppe Perinetti, Eugenio Pedullà, Antonio Scarano, Claudio Stacchi

**Affiliations:** 1Independent Researcher, 73100 Lecce, Italy; 2Department of Medical, Surgical and Health Sciences, University of Trieste, 34100 Trieste, Italy; fberton@units.it (F.B.); rapani.antonio@gmail.com (A.R.); rdilenarda@units.it (R.D.L.); gperinetti@units.it (G.P.); claudio@stacchi.it (C.S.); 3Department of General Surgery and Surgical-Medical Specialties, University of Catania, 95100 Catania, Italy; eugenio.pedulla@unict.it; 4Department of Medical, Oral and Biotechnological Sciences, University of Chieti-Pescara, 66100 Chieti, Italy; ascarano@unich.it

**Keywords:** piezoelectric surgery, endodonticsurgery, crack formation, ultrasonic tip, retropreparation

## Abstract

Although the application of ultrasounds in endodontic surgery allows for effective debridement of the root canal, incorrect device setting or inefficient tips seem to generate cracks during root-end retropreparation. The primary aim of this in vitro study was to establish the presence, or absence, of a correlation between ultrasonic root-end preparation and the formation of cracks. The present study was conducted on human teeth, extracted for periodontal reasons. After root canal treatment, roots were resected 3 mm from the anatomical apex by using a high-speed handpiece and carbide burs. The resected teeth were retroprepared by using an ultrasonic tip (R1D, Piezomed, W&H, Bürmoos, Austria), setting the piezoelectric device at maximum power available for the tip. Time required for the retropreparation was recorded. Before and after retropreparation, all roots were photographed under a stereomicroscope and analyzed by two different operators to evaluate: (a) the presence and extension of dentinal cracks and (b) the morphology of root-end preparation. Finally, piezoelectric tips were analyzed by scanning electron microscopy (SEM) to evaluate morphologic changes after use. A total of 43 single roots (33 with one root canal, 10 with two root canals) were treated. Average preparation time was 1 minute and 54 seconds. None of the roots without initial cracks developed new cracks after retropreparation. Quality of the preparation margins was fairly equal among the prepared specimens. None of the piezoelectric tips broke during instrumentation, and SEM analysis showed minimal surface wear of the tips after performing 11 retropreparations. Within the limits of the present study, the tested piezoelectric system does not seem to represent a major cause for root crack formation. Pre-existing cracks may expand after ultrasound root-end preparation.

## 1. Introduction

Ultrasounds were first introduced in endodontic surgery by Richman [1], with the aim of improving the effectiveness of root canal debridement and of performing both resection and retropreparation of the apical part of the dental root. Today, clinicians often choose ultrasonic root-end preparation, mainly because of the unmatched visibility this technology allows. This advantage is due to the angled shape of the tips, and to the cavitation effect, and allows to reduce the angle of the resection bevel [2,3,4,5,6,7,8]. Despite the excellent results obtained with the ultrasonic tips, some drawbacks have been associated with the use of this technique [9], including the presence of dentinal cracks on the resected root-end [10] and risk of perforation.

The contact between the instrument and root canal walls during preparation creates stress concentration in dentin and microcrack formation [11]. These microcracks are important because they may further develop into vertical root fractures. A recent study demonstrated that root fracture is not an instant event but rather a gradual propagation of tiny, less pronounced craze lines in the tooth structure [12].

In recent years, the occurrence of root fracture in either sound or endodontically treated/restored teeth has become a major concern in endodontics [13,14,15]. Some authors demonstrated that the endodontic procedures may increase the incidence of dentinal defects, such as Shemesh et al. [16] and Bier et al. [17]. Great interest was placed on the dentinal microcrack phenomenon by clinicians, academics and researchers over the following years. In a recent narrative review conducted on crack formation, Versiani et al. analyzed how the root dentinal microcracks observed in cross-sectional images of extracted teeth are not caused by canal-shaping procedures, and dehydration often causes cracking of the dentinal tissue, regardless of canal instrumentation [18].

In endodontic surgery, Layton et al. [19] suggested that ultrasonic root-end preparation might increase the risk of crack formation and found different types of cracks which they classified as follows:Intra-canal cracks start at the inner part of the canal and run through the dentine. They can be complete, if reaching the root surface, or incomplete, if ending inside the dentin.Intra-dentin cracks only affect the dentin, are usually distal or mesial to the canal and develop from buccal to lingual, and vice versa.Cement cracks start inside the cement and expand to the cement–dentin junction in a radial pattern.

The primary aim of this work was to investigate in vitro the influence of ultrasonic root-end preparation on the formation of different types of cracks. The ultrasonic tips used were evaluated by assessing the overall quality of the retrograde cavities and the effect of multiple uses on the tip itself.

## 2. Materials and Methods

This in vitro study investigated the integrity of human single roots after retrograde cavity preparation performed with a piezoelectric device. Quality and operative time of the preparations were evaluated, as well as the presence of cracks before and after ultrasonic instrumentation. Cracks were also recorded based on location and extension. Piezoelectric tips were examined after using scanning electron microscopy (SEM) to evaluate surface and shape alterations.

### 2.1. Specimen Selection

A total of 56 human teeth extracted for periodontal reasons from patients of 57 to 84 years old were cleaned from calculus and decay and stored in HBSS Solution (Hanks’ Balanced Salt Solution) at room temperature for a period of two to four weeks. A preliminary evaluation of the existence of fractures or dentinal cracks due to the extraction procedure was performed, using a microscope at 16× magnification (Leica 320, Leica Microsystems, Wetzlar, Germany). Teeth exhibiting radicular alterations or with incomplete or reabsorbed apices were discarded. A total of 33 single-rooted premolars and 10 mesial roots of mandibular molars were selected for treatment.

### 2.2. Specimen Preparation and Analysis

The crowns were resected to simplify the endodontic procedure and iconographic acquisition. All the teeth were endodontically treated following a crown-down approach. Canals were shaped to the working length with a rotary sequence (Protaper Universal, Dentsply Maillefer, Ballaigues, Switzerland) up to the F3 instrument. Canals were then obturated using warm vertical condensation [20] and sealer (Pulp Canal Sealer EWT™, Kerr Dental, Orange, CA, USA). Backpacking was performed by condensation of thermolasticized gutta-percha (Obtura III, Obtura Spartan, Algonquin, IL, USA).

All roots were resected 3 mm from the anatomical apex by using a high-speed handpiece with multiblade carbide bur (H847KRG314.016/018, Komet, Besigheim, Germany) under water spray. Each carbide bur was replaced after resecting ten roots. The resected roots were soaked in blue ink (Pelikan, Schindellegi, Switzerland) balanced with salt for 48 h, then rinsed, photographed and examined under an optical microscope (Leica MZ16, Leica Microsystems, Wetzlar, Germany) at 16× magnification to evaluate the presence of cracks prior to the retropreparation. 

### 2.3. Root-End Preparation

Both root resection and root-end preparation were performed by the same expert endodontist (C.B.) under microscope magnification (Leica M320, Leica Microsystems, Wetzlar, Germany). All the root-ends were prepared using a piezoelectric device (Piezomed, W&H, Bürmoos, Austria), set to power (40/100) as suggested by the manufacturer for the use of the dedicated tip (R1D, Piezomed, W&H, Bürmoos, Austria), under continuous saline irrigation. Each specimen was prepared following a standard protocol, with an up and down motion until creating a 3 mm deep preparation, measured by means of a periodontal probe. The tip was only activated when in contact with the tooth. Each tip was used to perform 11 retropreparations and then replaced. Cavities were then rinsed with 5 mL of saline solution to eliminate debris and remnants. 

### 2.4. Image Recording and Analysis

All specimens were photographed under 16 × magnification (Leica MZ16, Leica Microsystems, Wetzlar, Germany) after root-end resection and after retropreparation. The photographs were paired and coded by an independent assessor (C.S.) and then evaluated by two blinded assessors (F.B. and A.R.). Comparison of paired photographs determined presence, characteristics and time of occurrence of each crack.

### 2.5. Crack Evaluation

Crack evaluation was conducted and scored according to Abedi’s method [21], as follows: Roots with no cracks after root resection (before root-end preparation) and no cracks after root-end preparation;Roots with no cracks after root resection (before root-end preparation) that developed cracks after root-end preparation;Roots with cracks after root resection (before root-end preparation), which became longer or wider after root-end preparation, or that developed new cracks during root-end preparation.

Cracks were also classified as follows: Intracanal: cracks originating within the canal and extending into dentin;Intradentinal: cracks enclosed within the dentin and separate from the root surface and the canal;Extracanal: cracks originating at the root surface and extending into dentin;Communicating: cracks extending from root surface to the canal.

### 2.6. Retrograde Cavity Evaluation 

The quality of the root-end cavity margins was scored according to the degree of defects [22] as follows: (0) ideal preparation, no detectable defects; (1) imprint, a single visible defect, likely produced by the contact between the angulated portion of the tip and the cavity margin; (2) microchipped, ragged margin; (3) chipped, ragged margin together with defects likely caused by the tip bouncing off the root surface. 

### 2.7. Tip Analysis

A qualitative analysis of the effects of usage on tip shape and surface topography was performed by using scanning electron microscopy (FEG ESEM XL 30; FEI, Hillsboro, OR, USA). The entire sample was divided into four groups (3 groups of 11 teeth and 1 group of 10 teeth): images of the ultrasonic tip used in each group were captured at 35×, 100× and 200× magnification and compared by 2 different investigators (F.B. and A.R.) with the images of a brand-new tip. 

### 2.8. Working Time

The entire retropreparation procedure was timed with a professional stopwatch from the first contact of the tip to the root-end to the last passage of the retropreparation (HS-80TW-1EF, Casio, Shibuya, Japan).

These data were then elaborated separately for roots with one single canal and roots with two canals.

### 2.9. Statistical Analysis

Average mean crack between the assessors (as ordinal data) was calculated for both the PRE and POST time points and used to assess the significance of the difference between the time points by means of the Mann–Whitney U-test. Inter-rater repeatability was evaluated using the percentage of agreement and by both unweighted and linear-weighted kappa coefficients presented as mean (95% CI). The kappa coefficient ranges from 0 for no agreement to 1 for perfect agreement. The following standards for strength of agreement for the coefficient have been proposed: 0.01–0.20, slight; 0.21–0.40, fair; 0.41–0.60, moderate; 0.61–0.80, substantial; and >0.80 almost perfect [2]. Crack type was scored as follows: intracanal (1); intradentinal (2); extracanal (3) and communicating (4). Wilcoxon paired signed-rank test assessed the significance of the difference in the crack type between the ‘pre’ and ‘post’ root-end preparation. A p value less than 0.05 was used for the rejection of the null hypothesis.

## 3. Results

### 3.1. Examiners’ Agreement

Overall median (25th; 75th percentile) of the crack modality was 1.0 (0–3.0) and 2.3 (0–4.0) at the PRE and POST time points, respectively. The difference between the time points was not significant (*p* = 0.258 Mann–Whitney; p = 0.136 Wilcoxon).

The overall percentage of agreement between the raters was 72.7% (32 cases out of 44) for both the PRE and POST time point assessments, respectively (Table 1). For the PRE time point, unweighted and weighted kappa coefficients were 0.639 (0.466–0.811) and 0.700 (0.533–0.868), respectively. For the POST time point, unweighted and weighted kappa coefficients were 0.610 (0.437–0.783) and 0.741 (0.599–0.884), respectively.

### 3.2. Crack Presence and Evaluation

Of the 43 prepared roots, 34 were not affected by resection of the apex, while 9 roots showed the presence of cracks, namely 4 intracanal cracks, 2 intradentin cracks, 2 extracanal cracks and 1 communicating crack. After retropreparation, none of the sound roots showed newly formed cracks, while one intracanal crack was eliminated during retropreparation. The only communicating crack was unvaried after retropreparation. All the other cracks (i.e., 3 intracanal, 2 extracanal, 2 intradentin) turned into communicating cracks. Analysis of pre- and post-treatment crack type variation was reported in Table 2.

### 3.3. Quality of the Retrograde Cavity 

There was a total of 31 roots showing ideal preparation (0); 3 roots showing microchipping, ragged margin (2); 5 roots showing chipping (3); and 4 roots showing imprint (1) (Figure 1 and Figure 2).

### 3.4. Working Time

The working time was registered for the entire time for preparation of all specimens (total time 01:21:31). Mean root-end preparation time was 114.00 ± 69.32 seconds. In Table 3 are reported the data for single canals and double canals. 

### 3.5. SEM Evaluation of the Tips

Surface modifications of the R1D tips after 11 root-end preparations were minimal. Slight rounding of the diamond crystal edges was found, and very few crystals were lost during instrumentation (Figure 3). No relevant difference was found based on the working time of each tip. Tip 1 was used for 27 min 23 s, tip 2 for 18 min 58 s, tip 3 for 15 min 8 s, and tip 4 for 20 min 2 s. 

## 4. Discussion

The clinical outcomes of endodontic surgery have greatly improved in recent years, thanks to the adoption of microsurgical instruments, which have made management of the apical third [2,3,23] more efficient. To date, it is unknown if root-end alterations induced by retro-tips could affect the short and long-term clinical outcome, but any approach aimed at minimizing adverse effects (e.g., cracks) should be considered [22].

The present in vitro study was performed on human teeth extracted for periodontal reasons. Some teeth presented cracks prior to the root-end preparation, which could have been present prior to extraction or may have occurred during the extraction maneuvers or during the shaping procedures [18,24]. In fact, in vitro preparation may cause cracks more often than in vivo because of the shock-absorbing capacity of the periodontal ligament and because of the dehydration occurring during the shaping procedure [7,25]. In the present study, cracks were visible after root resection, while no cracks developed during ultrasonic root-end preparation. This result is in contrast to the supposed augmented risk of developing cracks upon ultrasonic root-end instrumentation. On the other hand, 77.8% (seven out of nine) of the present cracks were found to have worsened after preparation of the retrograde cavity. These results suggest that intact roots are at low risk of developing a crack. Existing cracks may extend or change in morphology. 

Few studies have investigated the different types of cracks produced after root-end preparation with ultrasonic retro-tips [24,25,26]. Rainwater et al. [24] found no significant difference in prevalence and type of crack when comparing a stainless-steel and a diamond retro-tip, the ultrasonic device set at low power. Beling et al. [26] found intradentinal and incomplete cracks after root-end preparation using a stainless-steel retro-tip, the ultrasonic device set at low power. 

Margin quality of the retrograde cavities does not seem to be affected by the power setting and the oscillations of the piezoelectric device, in agreement with other studies [10,27,28,29]. Moreover, tips were changed every 11 preparations to standardize the approach, but the operator did not notice a decrease in cutting efficacy, as verified upon SEM examination of the used tips which did not show significant signs of surface wear [30,31,32,33].

## 5. Conclusions

The present study showed encouraging results in retro-preparation performed with W&H Piezomed (W&H, Bürmoos, Austria). Although ultrasound root-end preparation did not cause any cracks, it seems that existing cracks might expand upon ultrasonic instrumentation [34].

## Figures and Tables

**Figure 1 bioengineering-09-00103-f001:**
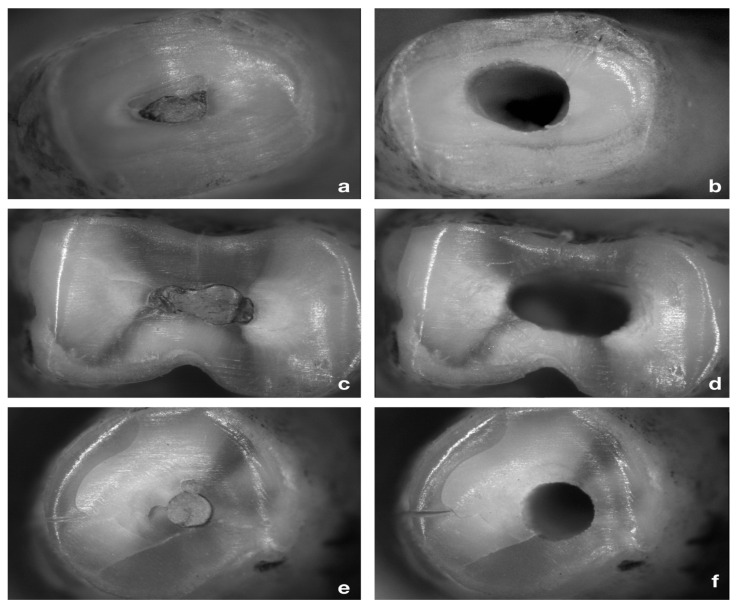
(**a**) Preoperative view of a single canal root; (**b**) Postoperative view of canal (**a**), absence of cracks; (**c**) Preoperative view of a mesial root; (**d**) Postoperative view of (**c**), note the precision of the preparation; (**e**) Preoperative view of a single canal root with the presence of cracks; (**f**) Postoperative view of (**e**), note the development of the crack.

**Figure 2 bioengineering-09-00103-f002:**
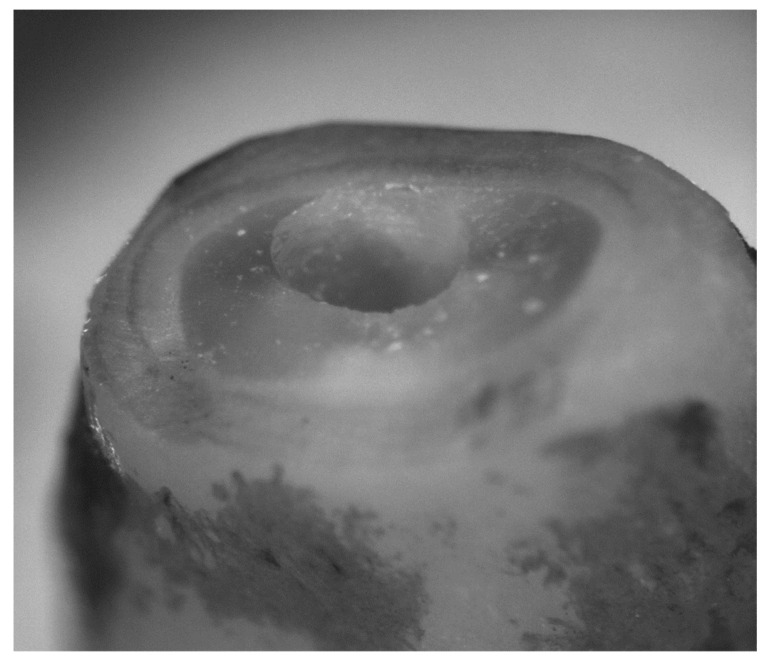
Details of the preparation.

**Figure 3 bioengineering-09-00103-f003:**
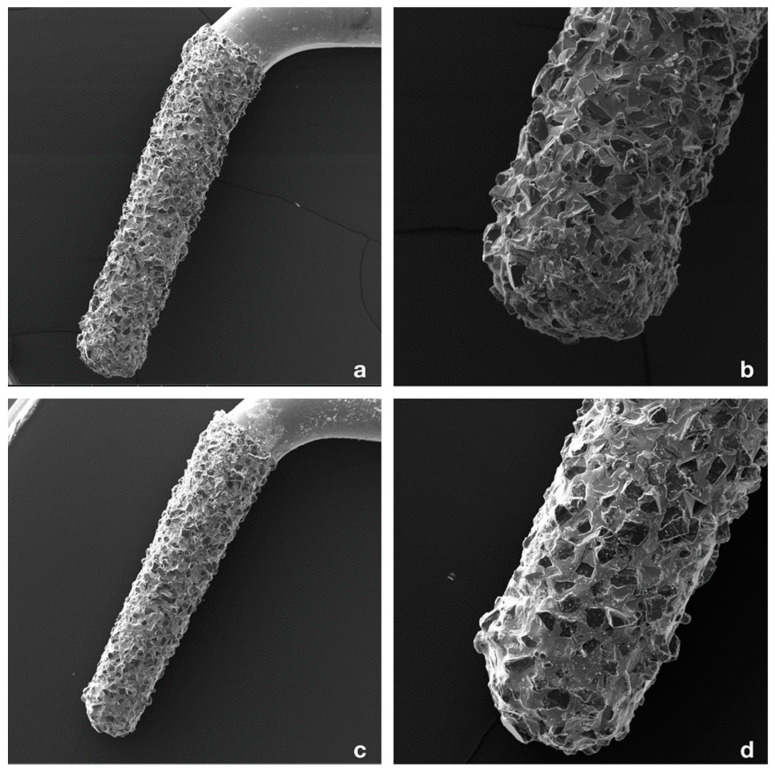
(**a**) SEM magnification of tip R1D (Piezomed, W&H, Bürmoos, Austria); (**b**) Details of the tip; note the regular position of the diamonds; (**c**) SEM magnification of tip R1D (Piezomed, W&H, Bürmoos, Austria) after utilization (20 min and 2 s); (**d**) Details of the tip; note the reduction of the number of diamonds compared to (**a**).

**Table 1 bioengineering-09-00103-t001:** Crosstabulation of the different crack modalities between the assessors according to the time points.

Time Point	Assessor FB	Assessor AR				
		None	Intracanal	Intradentinal	Extracanal	Communicating
Pre	None	12	1	0	0	0
	Intracanal	2	8	0	0	2
	Intradentinal	0	0	2	1	1
	Extracanal	0	0	0	3	1
	Communicating	0	3	0	1	7
Post	None	13	2	0	0	0
	Intracanal	1	4	0	2	2
	Intradentinal	0	0	0	0	2
	Extracanal	0	2	0	1	2
	Communicating	0	1	0	0	14

**Table 2 bioengineering-09-00103-t002:** Pre- and post-treatment crack type variation analyzed by Wilcoxon paired signed rank test.

Crack Type	Mean ± SD	Diff. *
Pre	0.42 ± 0.96	<0.05; S *
Post	0.74 ± 1.57	

* Diff.—significance of the difference; S—statistically significant.

**Table 3 bioengineering-09-00103-t003:** Time evaluation.

	Minutes	Seconds
mean	01:54	113.74
SD	01:09	69.32
mode	01:32	92.00
median	01:33	93.00
sd single canal	01:08	67.61
mode single canal	01:32	92.00
median single canal	01:32	93.00
mean double canal	02:23	142.6
sd single double canal	01:09	69.28
mode double canal	N/A *	N/A *
median double canal	02:44	164.00

* N/A—not applicable.

## Data Availability

The data supporting the findings of the present study are available from the corresponding author upon reasonable request.

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
