# Peer review of "In Vitro Qualitative Evaluation of Root-End Preparation Performed by Piezoelectric Instruments"

_bioengineering, 2022, doi:10.3390/bioengineering9030103_

Round 1

Reviewer 1 Report

The present manuscript aimed to evaluate the impact of root-end preparation on the formation of dentinal microcracks. For this, authors evaluated resected roots before and after retropreparation with ultrasonic tips. While the analitical tools are not the best available to evaluate dentinal microcracks formation (It would be better to use microCT technology), the manuscript might add some information for Endodontic literature. Prior my final decision some points should be addressed by the authors:  
  • First, authors should provide a better background, with extensive literature review, regarding the microcracks formation after root resection and retro-preparation in the Introduction section. 
  •  In item 2.3, when referring to the power used, I believe you need to make a small change in the text, so that it becomes clearer the power of the used ultrasonic device. It ranges from 0 to 100? This information is not clear.
  • Moreover, why the apicectomized roots were not soaked in blue ink after root-end preparation? This would facilitate the assessment of the root surface. Such protocol is widely used in clinics. Please discuss this item. 
  • If researchers have access to SEM, why not evaluate the roots using this tool? Please discuss this. 
  • Moreover, I suggest a discussion regarding the recent advances in microcracks detection, specially those manuscripts published by De-Deus et al. and Versiani et al. groups.

Author Response

We decided not to soak in the blue ink the samples after retropreparation because under stereomicroscope cracks were well visible, given the highly contrasted images obtainable with this device. In my own knowledge articles that suggest to soak in ink after retropreparation evaluate the microleakage of the filling materials. In this study we focused our attention only on the crack presence that resulted fully visible. 

We decide to analyze the samples only under stereomicroscope in order to avoid artifacts of the samples due to sample preparation and SEM evaluation. In fact, dehidratation of the sample, high vacuum and electron beam are proven to be able to alter noticeably hard matter such as dentinal samples.

Agee KL, Pashley EL, Itthagarun A, Sano H, Tay FR, Pashley DH. Submicron hiati in acid-etched dentin are artifacts of desiccation. Dent Mater. 2003 Jan;19(1):60-8. doi: 10.1016/s0109-5641(02)00007-6. PMID: 12498898.

Calzonetti KJ, Iwanowski T, Komorowski R, Friedman S. Ultrasonic root end cavity preparation assessed by an in situ impression technique. Oral Surg Oral Med Oral Pathol Oral Radiol Endod. 1998 Feb;85(2):210-5. doi: 10.1016/s1079-2104(98)90428-0. PMID: 9503458.

We added some bibliographic references about the micro CT analysis of dentinal microcracks. Literature show how this method is superior in cracks detection and not destructive, especially when used in fresh cadaver blocks.

The result of this study want to show how the use of the piezo  W&H is safe in retropreparation.  

Reviewer 2 Report

very nice paper with sound scientific methodology and excellent images.

I suggest to add the following citation:

Plotino G, Pameijer CH, Grande NM, Somma F. Ultrasonics in endodontics: a review of the literature. J Endod. 2007 Feb;33(2):81-95. doi: 10.1016/j.joen.2006.10.008. PMID: 17258622.

I cannot revise English language and style, so I leave this analysis to other referees or to the editor.

For what concerns the content, I recommend publication.

Author Response

Thank you  for your revision. We add the reference